# Copper(I)-catalysed site-selective C(sp³)–H bond chlorination of ketones, (E)-enones and alkylbenzenes by dichloramine-T

Jianwen Jin[1,5], Yichao Zhao[1,5], Sara Helen Kyne [1], Kaveh Farshadfar[2], Alireza Ariafard[2,3 ✉] & Philip Wai Hong Chan [1,4 ✉]

Strategies that enable intermolecular site-selective C–H bond functionalisation of organic molecules provide one of the cornerstones of modern chemical synthesis. In chloroalkane synthesis, such methods for intermolecular site-selective aliphatic C–H bond chlorination have, however, remained conspicuously rare. Here, we present a copper(I)-catalysed synthetic method for the efficient site-selective C(sp³)–H bond chlorination of ketones, (E)-enones and alkylbenzenes by dichloramine-T at room temperature. A key feature of the broad substrate scope is tolerance to unsaturation, which would normally pose an immense challenge in chemoselective aliphatic C–H bond functionalisation. By unlocking dichloramine-T's potential as a chlorine radical atom source, the product site-selectivities achieved are among the most selective in alkane functionalisation and should find widespread utility in chemical synthesis. This is exemplified by the late-stage site-selective modification of a number of natural products and bioactive compounds, and gram-scale preparation and formal synthesis of two drug molecules.

[1] School of Chemistry, Monash University, Clayton, Victoria, Australia. [2] Department of Chemistry, Islamic Azad University, Poonak, Tehran, Iran. [3] School of Natural Sciences–Chemistry, University of Tasmania, Hobart, Tasmania, Australia. [4] Department of Chemistry, University of Warwick, Coventry, UK. [5]These authors contributed equally: Jianwen Jin, Yichao Zhao. ✉email: alireza.ariafard@utas.edu.au; phil.chan@monash.edu

Organochlorides are ubiquitous in nature and can be found in nearly every class of biomolecule ranging from alkaloids to terpenoids to steroids with >2300 natural products containing the structural motif having been identified to date[1,2]. The compound family is also a versatile synthetic building block in chemical synthesis and a key component in a myriad of functional materials and active pharmaceutical ingredients[3–8]. For this reason, the development of new and efficient strategies that enable site-selective chlorination at a specific C–H bond in an organic compound continues to be an immensely important pursued field of research in chemical synthesis[3,9–36]. In recent years, this has been further driven by the prospect of such synthetic methods to streamline the assembly and modification of a broad spectrum of synthetically and therapeutically valuable targets[37–41]. Current chlorination protocols, for example, are a heavily relied upon synthetic tool in drug discovery programmes to improve the pharmacological and pharmacokinetic profiles of lead candidates[6,8–17].

In marked contrast to the number of elegant synthetic strategies for aromatic chlorination, those that enable site-selective intermolecular C($sp^3$)–Cl bond formation in aliphatic substrates are significantly less common (Fig. 1a)[9–17]. In the case of activated and benzylic C($sp^3$)–H bond chlorination, the majority are electrophilic or radical-based synthetic methods. The synthetic methods typically exploit chlorinating agents such as NCS (N-chlorosuccinimide) and NaOCl under acidic, basic, high temperature, or UV irradiation reaction conditions to achieve moderate to excellent product site-selectivities[9–12,18–23]. For allylic or unactivated C($sp^3$)–H bond chlorination, the analogous site-selective reactions are even fewer and continue to pose a formidable synthetic challenge for a number of reasons[24–36]. In allylic substrates, this is due to the propensity for electrophilic heterocycles or metal-oxo complexes, both widely used for aliphatic C–H bond functionalisation, to react with the more reactive alkene motif that results in C=C bond migration in the product[24–27,41]. The second is the often-associated difficulties of overcoming the increase in bond dissociation energies (BDE) on ongoing from a benzylic or allylic (81–87 kcal/mol) to an unactivated C($sp^3$)–H (92–99 kcal/mol) bond[42]. Added to this is the non-acidic nature of the covalent bond, with typical p$K_a$ values ≥50, and limited interactions with more polarisable and electronically accessible π-orbitals at an unactivated C($sp^3$)–H bond. Unlike alkane bromination, a further complication demonstrated in seminal works is often the less well controlled, predictable and regioselective outcome of functional group transformations involving elemental chlorine[43]. This poor site-selectivity and propensity to furnish polychlorinated adducts from more complex substrates is a consequence of the promiscuity of the chlorine-free radical atom. A recent notable advance to improving the site-selectivity of this approach, in this regard, has involved the development of N-chloro-based reagents such as the example shown in Fig. 1b for the chlorination of methyl hexanoate[21,36].

Another synthetic approach that has received attention is the realisation of a small handful of transition metal-catalysed synthetic strategies for intermolecular aliphatic C–H bond chlorination that avoid the intermediacy of the chlorine-free radical atom[11,28–35]. A recent illustrative example of this is the biomimetic-inspired manganese(III) porphyrin-catalysed C($sp^3$)–H bond chlorination of cyclobutane esters and bridged lactones with NaOCl (Fig. 1c)[28]. However, missing from this inchoate repertoire is an approach for the site-selective C($sp^3$)–H bond chlorination of ketones despite their ubiquity and the ease of synthesis of the functional group. In this context and as part of an ongoing programme focused on developing catalytic C–H bond functionalisation strategies, we were first drawn to the potential copper-mediated reactivity of ketones

with dichloramine-T (Fig. 1d)[44–47]. Our reaction design was inspired by a study showing the α-chlorination of bis(2-chloroethyl) sulfane by dichloramine-T via a proposed intermolecular hydrogen atom transfer (HAT) pathway[48]. In the intervening years, this mode of reactivity has been overlooked presumably due to a number of works establishing the N,N-dichloro reagent as a reliable electrophilic nitrogen source for transition metal-catalysed alkene amination, aziridination and chloroamination[49–53]. We were also guided by a computational study showing polar effects in carboxylic acids to deactivate the α-position, and to a lesser extent the β-position, in the alkyl side-chain toward HAT reaction with a chlorine radical atom[54,55].

Here, we describe the realisation of a CuOTf-catalysed synthetic method for the efficient site-selective chlorination at an unactivated secondary and tertiary γ-C($sp^3$)–H bond of a wide variety of ketones by dichloramine-T. The Cu(I)-mediated synthetic protocol is shown to additionally mediate the site- and chemoselective C($sp^3$)–H bond chlorination of a wide range of (E)-enones and alkylbenzenes by dichloramine-T. Achieved at room temperature, it delivers the corresponding γ-chloroketones, (E)-γ-chloroenones and α-benzyl chlorides as the only product. The practical utility of the Cu(I)-catalysed synthetic method is further demonstrated by showing the site-selective chlorination of a number of natural products and drug molecules. Along with this is the gram-scale preparation of a non-peptide δ-opioid agonist and formal synthesis of a melanin-concentrating hormone receptor 1 (MCHR1) antagonist[56,57]. Finally, we present mechanistic studies that reveal an intermolecular HAT pathway involving the Cu(II) complex **I** and [TsNCl]• motif shown in Fig. 1d as the two key active chlorination species and their role in determining the origin of the observed product selectivities.

## Results and discussion

**Reaction development**. Our studies began by examining the transition metal-catalysed C($sp^3$)–H bond chlorination of 4-methyl-1-phenylpentanone (**1a**) with dichloramine-T to determine the optimum reaction conditions (Table 1). This revealed that subjecting the ketone and chlorinating agent (1.5 equiv.) to 10 mol % of CuOTf, NaHCO$_3$ (1.5 equiv.) and 4 Å molecular sieves (MS) in acetonitrile at room temperature for 24 h gave the best result (entry 1). Under these reaction conditions, this provided 4-chloro-4-methyl-1-phenylpentanone (**2a**) in a 78% yield. The structure of the organochloride adduct was determined by comparison with the NMR measurements and X-ray crystallography of a closely related analogue vide infra. A survey of other copper catalysts showed the performance of Cu(OTf)$_2$ or Cu(OAc)$_2$ in place of CuOTf afforded comparable [1]H NMR yields of 74 and 72% (entries 7 and 8). Lower [1]H NMR yields of 50–65% were found on the repeating the reaction at a lower catalyst loading of 5 mol % of CuOTf or with CuOAc, Cu(TC), [Cu(MeCN)$_4$]PF$_6$, Cu(HETBA) or CuI as the catalyst (entries 2–6 and 9). Comparable [1]H NMR yields of 45 and 50% were observed on examining the transition metal complexes Rh$_2$(OAc)$_4$ or [Ir(COD)Cl]$_2$ instead of CuOTf as the catalyst (entries 19 and 21). In contrast, the analogous control reactions mediated by either Pd(OAc)$_2$ or FeCl$_3$ were observed to lead to the recovery of the substrate in near quantitative yield (entries 18 and 20). Similarly, control experiments in the absence of the catalyst, or mediated by CuOTf and with NaOAc or Et$_3$N instead of NaHCO$_3$ as the base, were found to lead to the recovery of the substrate in near quantitative yield (entries 10, 15 and 16). On the other hand, the analogous CuOTf-catalysed experiments at a higher reaction concentration of 0.3 or 0.4 M, or where NaHCO$_3$ was absent or replaced by Na$_2$CO$_3$ or K$_3$PO$_4$ as the base were shown to provide [1]H NMR yields of 51–80% (entries 11–14 and 17).

**Fig. 1 Synthetic strategies for intermolecular site-selective C–H bond chlorination. a** A comparison of the number of synthetic methods for site-selective aromatic, benzylic, allylic and alkyl chlorination. **b** Example of site-selective aliphatic chlorination by *N*-chloro-based reagents. **c** Example of site-selective aliphatic chlorination by directing group-assisted transition metal catalysis. TPFPP = 5,10,15,20-tetrakis(pentafluorophenyl)porphyrinate. **d** Cu(I)-catalysed site-selective C(*sp*³)–H bond chlorination of ketones, (*E*)-enones and alkylbenzenes by dichloramine-T.

In addition to the control experiments described in Table 1, the CuOTf-catalysed chlorination of **1a** by dichloramine-T under the various reaction conditions described in Supplementary Table 1 were examined. Along with NaHCO₃ (1.5 equiv.) and 4 Å MS in acetonitrile at room temperature for 24 h, this showed the introduction of 10 mol % of 2,2′-bipyridine, 1,10-phenanthroline, (3a*S*, 8a*R*)-in-pybox or Xantphos as a ligand to the reaction conditions led to the recovery of only the substrate (entries 1–4). A similar outcome was observed on changing the chlorinating agent from dichloramine-T to chloramine-T, NaOCl, NCS, DCDMH (1,3-dichloro-5,5-dimethylhydantoin) or TCCA (trichloroisocyanuric acid) (entries 5–9). In a final set of control reactions, an evaluation of different reaction solvents showed control experiments with acetonitrile replaced by methanol, tetrahydrofuran, 1,3-dioxane, nitromethane, dichloromethane, or toluene also resulted in no reaction (entries 10–15). The only

exceptions were the analogous control experiments with benzonitrile as the solvent, or in acetonitrile in the absence of 4 Å MS, which gave ¹H NMR yields of 41 and 48% (entries 16 and 17).

**Substrate scope and utility**. To determine the generality of the present site-selective chlorination procedure, the reactions of a series of ketones containing a secondary or tertiary γ-carbon centre were first evaluated, and the results are summarised in Fig. 2a. Overall, these experiments demonstrated the CuOTf-catalysed reaction conditions to be broad, providing a variety of γ-chlorooketone derivatives in 40–83% yield from the corresponding carbonyl compounds **1b–x**. Reactions of substrates containing a phenyl motif with electron-donating (**1b**, **1c**, **1f** and **1g**) or -withdrawing (**1d**, **1e**, **1h** and **1i**) groups at various positions of the ring or phenanthrenyl (**1j**) group were found to proceed well. In these experiments, the corresponding γ-chloroketones **2b–j** were

## Table 1 Optimisation of the reaction conditions.

| Entry | [M] | Base | Yield (%)[a] |
|---|---|---|---|
| 1 | CuOTf | NaHCO$_3$ | 81 (78)[b] |
| 2 | CuOTf[c] | NaHCO$_3$ | 65 |
| 3 | CuOAc | NaHCO$_3$ | 65 |
| 4 | Cu(TC) | NaHCO$_3$ | 50 |
| 5 | Cu[(MeCN)$_4$]PF$_6$ | NaHCO$_3$ | 67 |
| 6 | CuI | NaHCO$_3$ | 64 |
| 7 | CuOTf$_2$ | NaHCO$_3$ | 74 |
| 8 | CuOAc$_2$ | NaHCO$_3$ | 72 |
| 9 | Cu(HETBA) | NaHCO$_3$ | 64 |
| 10 | –[d] | NaHCO$_3$ | –[e] |
| 11 | CuOTf | K$_3$PO$_4$ | 76 |
| 12 | CuOTf | Na$_2$CO$_3$ | 75 |
| 13[f] | CuOTf | NaHCO$_3$ | 80 |
| 14[g] | CuOTf | NaHCO$_3$ | 51 |
| 15 | CuOTf | NaOAc | –[e] |
| 16 | CuOTf | Et$_3$N | –[e] |
| 17 | CuOTf | –[h] | 68 |
| 18 | FeCl$_3$ | NaHCO$_3$ | –[e] |
| 19 | Rh$_2$(OAc)$_4$ | NaHCO$_3$ | 45 |
| 20 | Pd(OAc)$_2$ | NaHCO$_3$ | –[e] |
| 21 | [Ir(COD)Cl]$_2$ | NaHCO$_3$ | 50 |

All reactions were performed with **1a** (35.3 mg, 0.2 mmol) and TsNCl$_2$ (72 mg, 0.3 mmol) in the presence of the metal complex (0.02 mmol), 4 Å MS (50 mg), base (0.3 mmol) in acetonitrile (1 mL, 0.2 M) at room temperature for 24 h.
COD 1,5-cyclooctadiene, HETBA ethyl benzoylacetate, TC thiophene-2-carboxylate.
[a]Crude yield of **2a** as determined by [1]H NMR measurements with dibromomethane as the internal standard.
[b]Isolated product yield.
[c]Reaction performed at a catalyst loading of 5 mol %.
[d]Reaction performed in the absence of a catalyst.
[e]No reaction detected by TLC and [1]H NMR analysis of the crude reaction mixture.
[f]Reaction performed in 0.75 mL (0.3 M) of acetonitrile.
[g]Reaction performed in 0.5 mL (0.4 M) of acetonitrile.
[h]Reaction performed in the absence of a base.

furnished in yields of 51–83% with the structure of **2d** confirmed by X-ray crystallography (Supplementary Fig. 10). The presence of other branched alkyl (**1k–m**) or cycloalkyl (**1n** and **1o**) tethers of phenyl ketones were likewise tolerated, giving the corresponding γ-chlorination products **2k–o** in 40–79% yield. Similarly, reactions involving dialkyl-substituted substrates (**1p–r**) were found to furnish the corresponding γ-chloroketones **2p–r** in 59–76% yield. The reactions of substrates containing an alkyl tether with a secondary γ-carbon centre (**1s–x**) were observed to be the only exception. In these experiments, the further addition of 10 mol % of the catalyst and 1.5 equiv. of dichloramine-T after 24 h to give a total reaction time of 48 h, was needed to afford the corresponding γ-chloroketones **2s–x** in 45–78% yield. Beyond ketone substrates, we were pleased to find the site-selective Cu(I)-mediated chlorination protocol to proceed well with the carboxylic esters **1y** and **1z**, and amide **1α** to give the γ-chloroesters **2y** and **2z**, and γ-chloroamide **2α** in 40–50% yield. The chlorination of methyl hexanoate **1β** was also attempted but found to give a mixture of unidentifiable decomposition products based on GC and [1]H NMR measurements of the crude reaction mixture.

We next sought to assess the scope of the site-selective catalytic C(sp3)–H bond chlorination protocol with a range of (E)-enone and alkylbenzene substrates (Fig. 2b, c). This revealed experiments of α,β-unsaturated compounds with phenyl moieties containing various electron-donating (**3b**, **3c** and **3f**) or

-withdrawing groups (**3d** and **3e**) were found to be well tolerated (Fig. 2b). Under the optimised CuOTf-mediated reaction conditions, the corresponding (E)-allyl chlorides **4b–f** were obtained in yields of 63–83%. Likewise, substrates with a pendant 2-naphthalenyl (**3g**) or 2-thienyl (**3h**) group in place of the phenyl moiety were shown to proceed well, delivering the corresponding γ-chlorination products **4g** and **4h** in 38 and 37% yield, respectively. Reactions of (E)-phenylvinyl ketones containing a gem-dialkyl (**3a**, **3i** and **3j**) or cyclohexyl (**3k**) motif were similarly found to afford (E)-γ-chloroenones **4a** and **4i–k** in 62–89% yield with the structure of **4k** confirmed by X-ray crystallography (Supplementary Fig. 11). We were equally pleased to find the reaction of toluene (**5a**) and derivatives of the compound (**5b–d**) gave the corresponding α-chlorinated adducts (**6a–d**) in 52–71% yield (Fig. 2c). A similar outcome was found for the transformations of alkylbenzenes containing a p-ketone, -ester or -Weinreb amide moiety, as in **5e–j**, as well as α-indanone (**5k**) and α-tetralone (**5l**). In our hands, these reactions furnished the corresponding α-aryl chloride products (**6e–l**) in 60–90% yield. The assembly of **6f** is particularly noteworthy as it is a reported key intermediate in the synthesis of a potent MCHR1 antagonist[57]. Its preparation by the present method thus represents a formal synthesis of the bioactive compound. Added to this is the gram-scale synthesis of **6e** and its further transformation to the non-peptide δ-opioid agonist **7** in 96% yield when treated with N-Boc-piperazine under base-mediated reaction conditions[56].

The late-stage modification of natural products and drug molecules represent an efficient strategy for the facile synthesis of compounds with natural-product-like core structures in phenotype-based drug discovery programmes. With this in mind and as part of the study to define the synthetic utility of the present site-selective Cu(I)-catalysed C(sp3)–H bond chlorination procedure, its application to the late-stage modification of a variety of targets illustrated in Fig. 2d were examined. This showed that we were able to deconstruct and site-selectively chlorinate a member of the 4-pyridyl ketone family of tetrameric carbonyl reductase inhibitors, L-leucine, ibuprofen and the retinoic acid receptor morphogen retinoic acid to give the anticipated chlorinated products **8–11** in 50–85% yield. Equally, the analogous reactions with the antimicrobial and antifungal ligand camphanic acid, mixed dyslipidemia suppressant fenofibric acid and musk fragrance celestolide were found to produce the corresponding chloride derivatives **12–14** in 40–97% yield. Considered together, all the above-examined chlorination reactions demonstrated that the C(sp3)–Cl bond-forming process occurs in a highly selective manner. No other regioisomers or di- or trichlorinated adducts of the product resulting from an unselective or successive C(sp3)–Cl bond formation pathway were detected by [1]H NMR analysis of the crude reaction mixtures. As highlighted in Fig. 2a, b, this notably included ketones **1b**, **1c**, **1f**, **1g**, **1p**, **1q**, **1w** and **1x**, and (E)-enones **3b**, **3c**, **3f** and **3h** containing either or both a benzylic and electron-rich aryl C–H bond[9–23]. In the case of experiments involving (E)-enones, the possible formation of compounds due to competitive alkene amination, aziridination, chloroamination or chlorination were likewise not observed by [1]H NMR analysis of the crude reaction mixtures.

**Mechanistic studies**. To gain a better understanding of the possible mechanism of the Cu(I)-catalysed chlorination reaction, the following control experiments were performed (Fig. 3). In the first set of control experiments, subjecting **1a** and a stoichiometric amount of TEMPO or BHT to the CuOTf-catalysed reaction conditions depicted in Fig. 3a was found, in both cases, to lead to

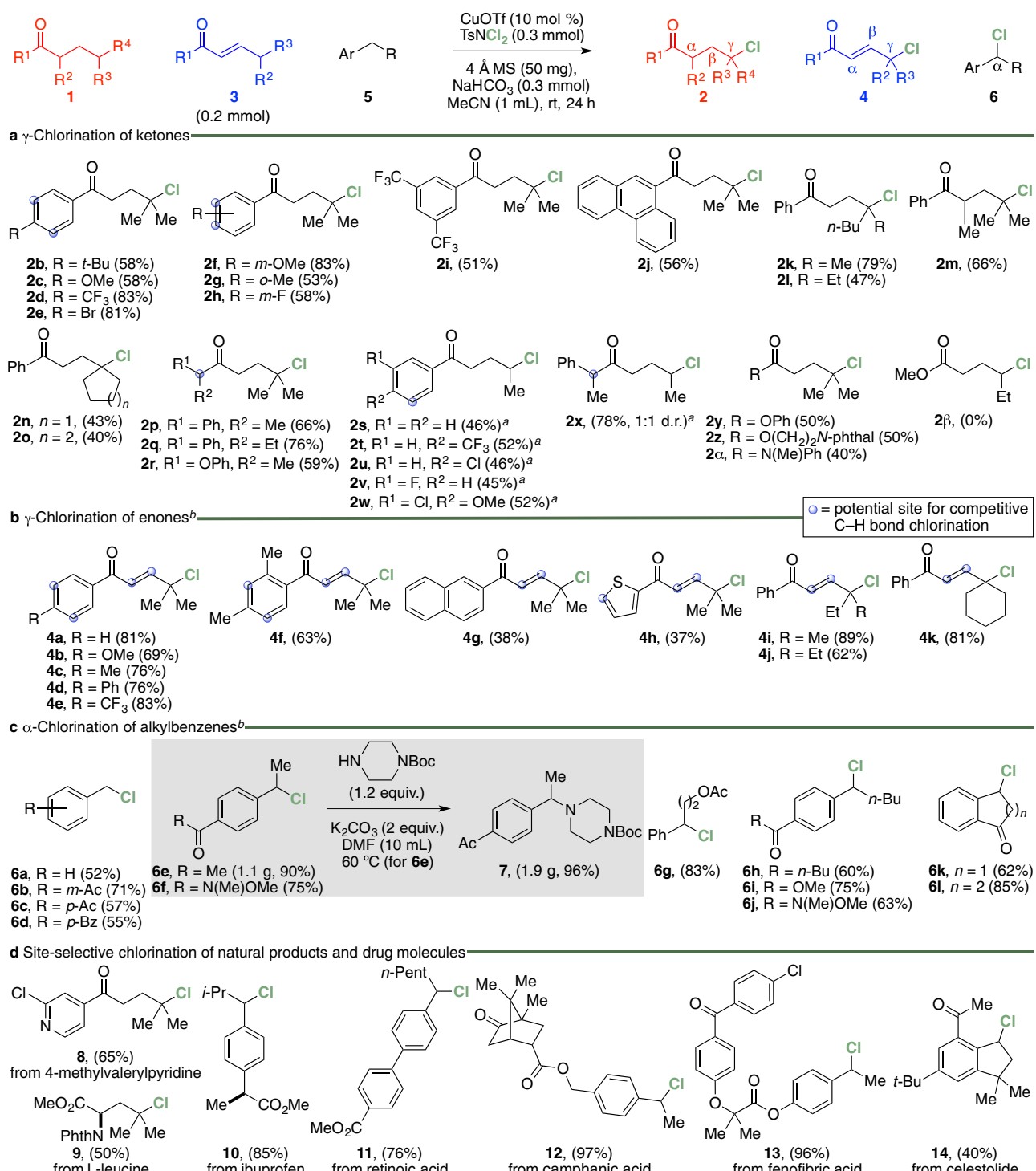

**Fig. 2 Scope and synthetic applications of the CuOTf-catalysed site-selective C(sp³)–H bond chlorination reaction. a** At the γ-position of ketones. **b** At the γ-position of (E)-enones. **c** At the α-position of alkylbenzenes. **d** Late-stage chlorination of natural products and drug molecules. [a]Reaction performed with 20 mol % of CuOTf and 0.6 mmol of dichloramine-T for 48 h. [b]Reaction performed with 0.24 mmol of dichloramine-T.

the recovery of the substrate in near quantitative yield. Suggesting the possibility of a radical pathway, this was further corroborated by a CuOTf-catalysed radical clock experiment with the cyclopropyl-substituted (E)-enone **3l**, which was found to result in complete substrate consumption (Fig. 3b). While the anticipated 1,3-chlorodiene adduct **15** could not be isolated by flash column chromatography as it was found to be too unstable, its presence in the crude reaction mixture was detected by HRMS measurements. Despite the mechanism for the cyclopropyl

ring-opening of **3l** currently remaining unclear, one possibility could involve a γ-H• atom abstraction/vinylcyclopropane-type rearrangement/dichlorination/dehydrohalogenation cascade to provide 1,3-chlorodiene **15**. The role of the carbonyl functionality in the ketone substrate in facilitating deactivation of the α- and β-positions of the alkyl side-chain was shown by control experiments with the ketone **1γ**, alkylbenzene **5m** and alcohol **17** (Fig. 3c–e). Under the CuOTf-catalysed optimised reaction conditions, this led to either the recovery of the substrate in near

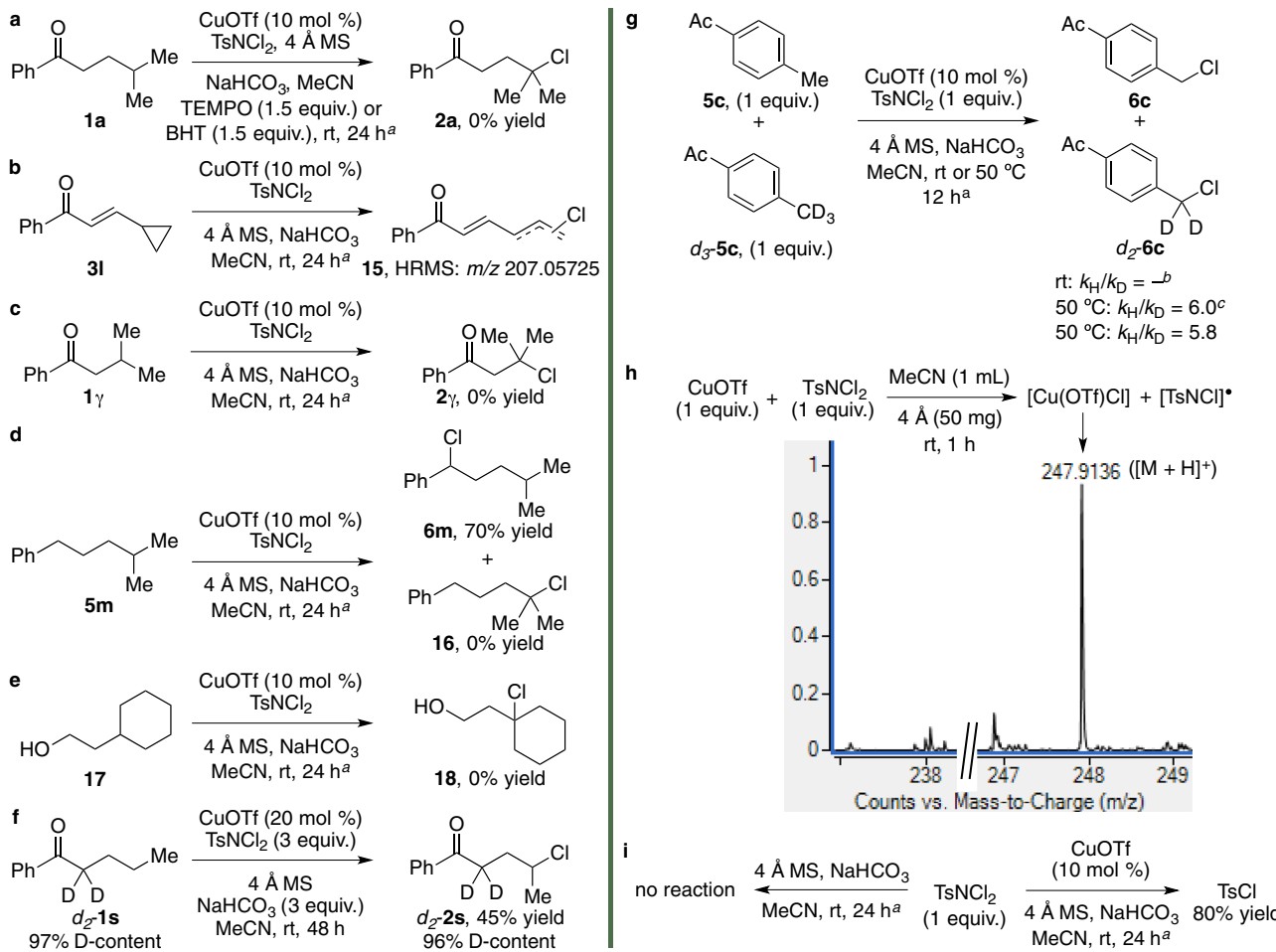

**Fig. 3 Control experiments. a** Influence of the radical scavengers TEMPO and BHT on the reaction of **1a**. **b** Radical clock experiment with **3l**.
**c–e** Determining the role of the carbonyl functionality in the substrate with reactions of **1γ**, **5m** and **17**. **f** Determining the possibility of an α-HAT reaction/
1,3-H• atom shift pathway with the reaction of $d_2$-**1s**. **g** KIE measurements with **5c** and $d_3$-**5c**. **h, i** Detection of the possible copper species generated from
the reaction of CuOTf and dichloramine-T by HRMS measurements and reaction products obtained. $^a$Unless otherwise stated, control reactions were
performed with 1.5 equiv. of dichloramine-T, 50 mg of 4 Å MS and 1.5 equiv. of $NaHCO_3$ in acetonitrile (1 mL). $^b$In parallel and competitive KIE experiments
at room temperature both gave **6c** as the only product in 32% yield. $^c$In parallel KIE experiments at 50 °C afforded **6c** and $d_2$-**6c** in 36 and 6% yield.

quantitative yield or the α-aryl chloride **6m** in 70% yield for the reaction with **5m**. These results also hinted that the lower BDE value of a tertiary (96.1 kcal/mol) over that of a secondary (99.1 kcal/mol) $C(sp^3)$–H bond was unlikely to be the sole contributing factor in determining site-selectivity[42]. The possibility of the substrate undergoing a α-HAT reaction/1,3-H• atom shift pathway was next considered but discounted by examining the CuOTf-catalysed chlorination of $d_2$-**1s** under the optimised reaction conditions for 48 h (Fig. 3f). This test gave the deuterated γ-chloroketone $d_2$-**2s** in 45% yield and with near-complete retention of D-content at the α-position of the product based on $^1$H NMR measurements. In a final set of control experiments, the in parallel and competitive deuterium kinetic isotope effect (KIE) of the present Cu(I)-catalysed chlorination protocol was examined with **5c** and $d_3$-**5c** as the test substrates (Fig. 3g). In both KIE experiments, analysis by $^1$H NMR and HRMS measurements revealed only the formation of **6c** in 32% yield along with the recovery of $d_3$-**5c** in near quantitative yield that implied the possible involvement of quantum mechanical tunnelling. Common in H• atom transfer pathways, this was further suggested by $k_H/k_D$ values of 6.0 and 5.8 obtained on repeating the respective in parallel and competitive KIE reactions at 50 °C. The theoretical

limit for $k_H/k_D$ (at 298 K) of 6.5–7 is estimated for complete C–H bond breaking in H• atom transfer reactions (ref. [58] and references cited therein). A large KIE value, like that measured in our study, is considered an experimental indicator that quantum mechanical tunnelling may be occurring (ref. [59] and references cited therein). This is further supported by our computational findings showing that the KIE value for H• atom abstraction from **5c** is 6.4, which is close to the theoretical limit and corresponds with our experimentally observed results. A HRMS measurement of the crude reaction mixture obtained from a control experiment with stoichiometric amounts of CuOTf and dichloramine-T in acetonitrile at room temperature for 1 h was likewise performed (Fig. 3h). This revealed the detection of a molecular ion at $m/z$ 247.9136 ([M+H]$^+$) that could be assigned to [Cu(OTf)Cl], which suggested this Cu(II) species could be formed in situ and be involved in the reaction pathway. Further support for the possible involvement of [Cu(OTf)Cl] was offered by the detection of no reaction and isolation of TsCl in 80% yield on repeating the control experiment in the absence of the catalyst or 10 mol % of CuOTf, respectively (Fig. 3i).

To gain further insight into the mechanism of the present site-selective CuOTf-catalysed chlorination protocol, attention was

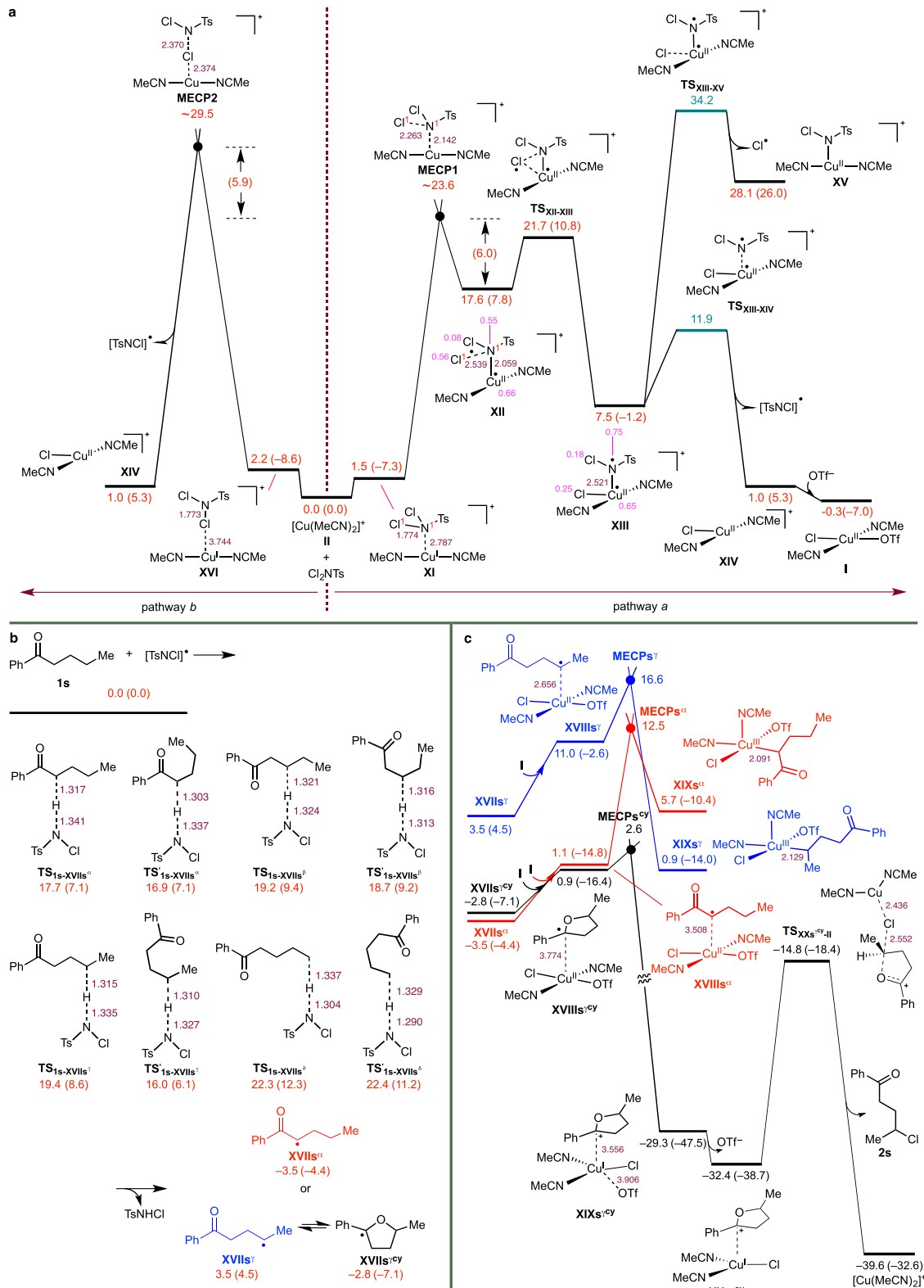

**Fig. 4 DFT calculations. a** Energy profiles for the oxidative addition of dichloramine-T to [Cu(CH₃CN)₂]⁺ **II** via the SET mechanistic pathways *a* and *b*.
**b** Energy barriers for the HAT reaction of [TsNCl]• at different positions of **1s** and relative energies of the ensuing radical species **XVIIs**$^α$, and **XVIIs**$^γ$ and **XVIIs**$^{γcy}$ produced from the two most favoured transition states, **TS'₁ₛ₋ₓᵥᵢᵢₛα** and **TS'₁ₛ₋ₓᵥᵢᵢₛγ**. **c** Energy profile for the chlorination of **XVIIs**$^α$, **XVIIs**$^γ$ and **XVIIs**$^{γcy}$ to **2s**. The relative Gibbs free and potential energies (in parentheses) obtained from SMD/B3LYP-D3/def2-TZVP//SMD/B3LYP-D3/6-31G(*d,p*), SDD calculations are given in kcal/mol (red), bond lengths in Å (purple) and spin density distribution values in e/Å³ (pink).

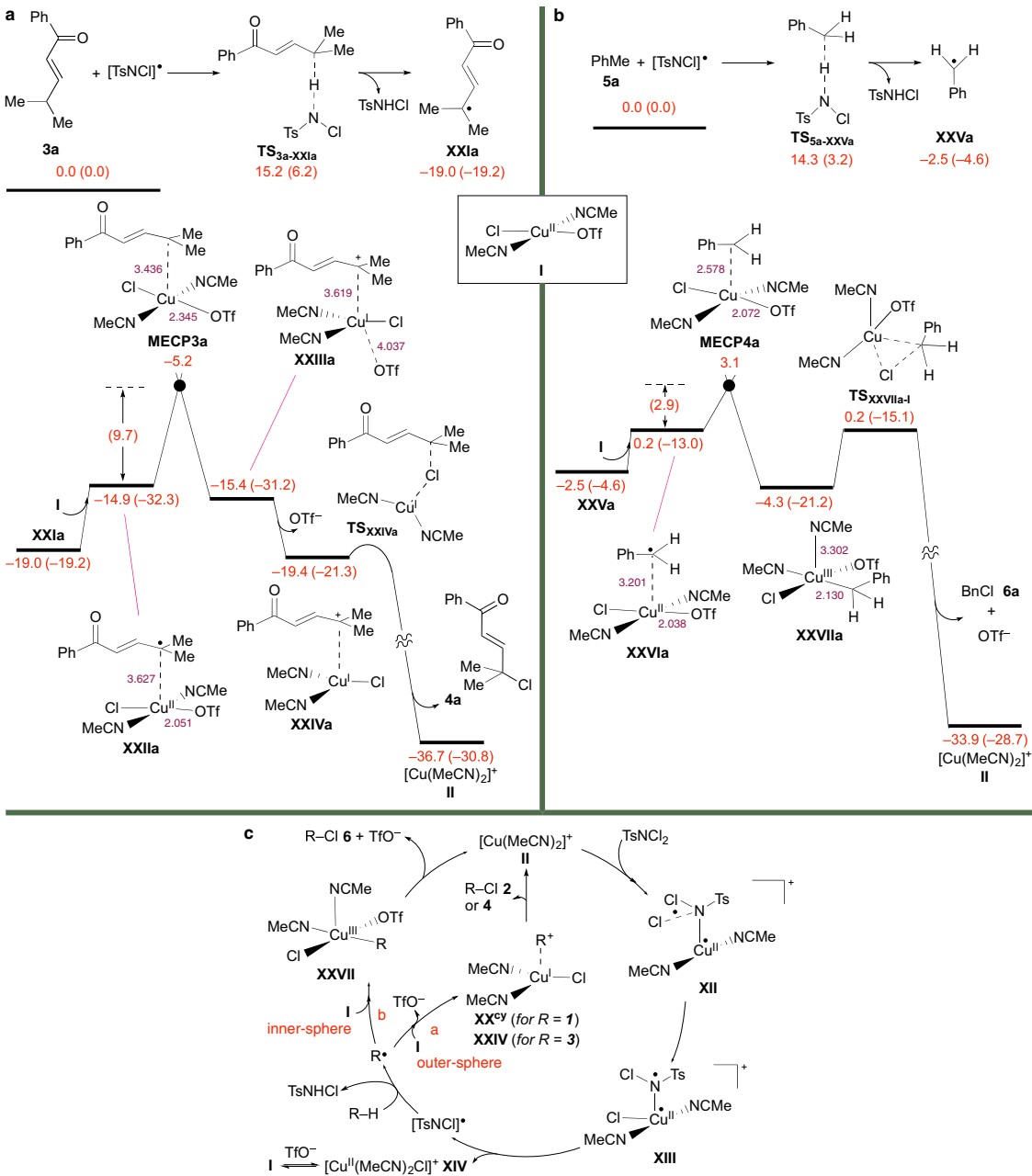

**Fig. 5 DFT calculations and proposed reaction mechanism.** Energy profiles for the chlorination of **a XXIa** to **4a** from **3a** and [TsNCI]• motif; and **b XXVa** to **6a** from **5a** and [TsNCI]• motif. The relative Gibbs free energies and potential energies (in parentheses) obtained from SMD/ B3LYP-D3/def2-TZVP//SMD/B3LYP-D3/6-31G(*d,p*),SDD calculations are given in kcal/mol (red) and bond lengths in Å (purple). **c** Proposed catalytic cycle.

next turned to undertaking a series of DFT (density functional theory) calculations (Figs. 4, 5 and Supplementary Figs. 12–23). With the reaction surmised to begin with the oxidative addition of dichloramine-T to the Cu(I) salt, the identity of the reactive metal species involved in this step was first examined (Supplementary Fig. 12). Performed at the SMD/B3LYP-D3/def2-TZVP// SMD/B3LYP-D3/6–31 G(*d,p*),SDD level of theory, a number of copper complex structures were considered and this revealed singlet ground-state [Cu(MeCN)₂]⁺ **II**, arising from coordination of the metal salt to two molecules of the solvent, to be the most stable. To this end, the cationic metal complex **II** was set as the reactive catalytic species for all subsequent calculations directed toward scrutinising whether this crucial oxidative addition step

proceeded via a single-electron transfer (SET) or concerted mechanism (Fig. 4a and Supplementary Fig. 13).

In considering a SET mechanism, our calculations characterised the possibility of two distinct pathways (Fig. 4a). The first involves dichloramine-T approaching the metal centre of **II** through its *N*-donor atom to give the T-shaped Cu(I)-encounter complex **XI** with an energy of 1.5 kcal/mol (Fig. 4a, pathway *a*). A feature of the organometallic species was found to be a Cu–N¹ bond distance of 2.787 Å due to the weak coordination of the metal centre to the nitrogen atom. The shortening of the Cu–N¹ bond concomitant with a lengthening of the N¹–Cl¹ bond predisposes an electron from the metal centre to be promoted to the N¹–Cl¹ bond σ* orbital via minimum energy crossing point

species **MECP1**. Providing the triplet excited-state copper(II) species **XII**, this type of bond rearrangement causes the $N^1$–$Cl^1$ bond to become weaker and more available to accept an electron from the copper centre. The occupation of the $N^1$–$Cl^1$ bond $\sigma^*$ orbital by an electron is supported by the elongation of the $N^1$–$Cl^1$ bond in **XII** and respective spin density distribution values of 0.66, 0.56, and 0.55 e/$Å^3$ for the Cu, $N^1$, and $Cl^1$ atoms in the complex (noted in pink in Fig. 4a). Once this copper(II) species has formed, the oxidative addition process was found to be completed by transferring the $Cl^1$ atom from the $N^1$ centre to the metal via transition structure **TS$_{XII-XIII}$** with an energy barrier of 4.1 kcal/mol. In the ensuing Cu(II) species **XIII** on the triplet surface, the [TsNCl]• motif weakly coordinates to the metal centre, as evidenced by a long Cu–$N^1$ bond distance of 2.521 Å when compared to that in **XII** (2.059 Å). The weakening of the Cu–$N^1$ bond thus enables the [TsNCl]• motif to easily dissociate from the metal complex. Calculated to be exergonic by 6.5 kcal/mol with an energy barrier of 4.4 kcal/mol, this generates the three-coordinate copper(II) complex **XIV** via **TS$_{XIII-XIV}$**. The newly formed Cu(II) complex can be further stabilised through its coordination with a triflate anion. This results in the formation of the four-coordinate Cu(II) species **I** with a relative free energy of 1.3 kcal/mol. Other Cu(II) complexes were also considered but discounted as they were found to be less stable than that of **I** (Supplementary Fig. 14). The competitive release of the Cl• atom instead of the [TsNCl]• group from copper(II) species **XIII** was likewise examined. However, this alternative pathway, which would produce the three-coordinate copper(II) complex **XV** via **TS$_{XIII-XV}$**, was calculated to be endergonic by 20.6 kcal/mol, with an activation barrier of 26.7 kcal/mol. These results, therefore, indicate that it is more favourable for the Cu(II) species **XIII** to liberate the [TsNCl]• motif than the Cl• atom.

A second possible SET mechanism that commences with the Cl-donor atom of dichloramine-T approaching the metal centre of [Cu(MeCN)$_2$]$^+$ **II** was next explored (Fig. 4a, pathway *b*). Similar to that observed in Fig. 4a, pathway *a*, this furnishes the T-shaped Cu(I)-encounter complex **XVI** with a free energy of 2.2 kcal/mol. Electron transfer in this case proceeded by surmounting the minimum energy crossing point species **MECP2**, which lies 5.9 kcal/mol higher in energy than **MECP1**, to give **I** and the [TsNCl]• motif. Consequently, this suggested dichloramine-T was a more potent oxidant if it interacted with the metal salt through its nitrogen centre.

In a third set of calculations, the possibility of oxidative addition of dichloramine-T to the active copper(I) species **II** proceeding by a concerted mechanism was investigated (Supplementary Fig. 13). This reveals the oxidation of Cu(I) species **XI** to Cu(III) complex **XIII$^s$** via transition structure **TS$^s_{XI-XIII}$** was endergonic by 20.6 kcal/mol with an overall activation barrier of 27.9 kcal/mol. The high endergonicity obtained for **XIII$^s$** is indicative of the fact that if the [TsNCl]• motif was formed, it would be unlikely to re-interact with a Cu(II) species on the singlet excited-state energy surface to give a stable Cu(III) complex. Taken together, these calculations thus suggests that the oxidative addition of dichloramine-T to [Cu(MeCN)$_2$]$^+$ **II** was most likely to proceed via the SET mechanism outlined in Fig. 4a, pathway *a*, as this was the most energetically favoured. It also suggests that the [Cu(MeCN)$_2$(OTf)Cl] **I** and [TsNCl]• moieties formed as a result of this step are the two key active species responsible for substrate chlorination. Corroborating this is our findings that showed [Cu(OTf)Cl] to be present in HRMS measurements of the crude reaction mixture and the isolation of TsCl in control experiments with CuOTf and dichloramine-T (Fig. 3h, i). Thus, the Cu(II) complex **I** and [TsNCl]• motif were set as the reactive chlorination species for all subsequent calculations illustrated in Figs. 4b, c and 5, which focused on

revealing the mechanisms of the site-selective C($sp^3$)–Cl bond formation reactions.

With this in mind and 1-phenylpentanone **1s** as a representative example, the free energy barriers for a HAT reaction at the $\alpha$-, $\beta$-, $\gamma$- and $\delta$-positions of the substrate with the [TsNCl]• moiety were initially investigated (Fig. 4b). This predicted the HAT reaction was most likely to proceed at the $\gamma$-position of the ketone with the alkyl side-chain occupying a folded conformation. The energy barrier for **TS$'_{1s-XVIIs}$$^\gamma$** formation was shown to be 16.0 kcal/mol, some 0.9 kcal/mol lower than that of **TS$'_{1s-XVIIs}$$^\alpha$**, the next lowest, which is obtained from HAT reaction at the $\alpha$-position of the substrate. This is consistent with the control experiment described in Fig. 3f, which revealed the chlorination of $d_2$-**1s** to give $d_2$-**2s** in 45% yield with complete D-content retention at the $\alpha$-position of the product. It is also in good agreement with computational calculations that reported polar effects to deactivate the $\alpha$-position of carboxylic acids to participate in a HAT reaction with a chlorine radical atom despite this being the most thermodynamically favoured[54].

Our theoretical calculations found that once the $\gamma$-carbon-centred radical species **XVIIs$^\gamma$** forms along with *p*-TsNHCl, it cyclises to the tetrahydrofuranyl (THFyl) radical adduct **XVIIs$^{\gamma c\gamma}$**, as it was more stable by 6.3 kcal/mol (Fig. 4b). Additionally, a comparison of the cyclic carbon-centred radical species along with that of **XVIIs$^\alpha$** and **XVIIs$^\gamma$** showed the odd-electron THFyl adduct to exhibit significantly greater reactivity toward chlorination. We rationalise this marked difference in reactivity could offer the reason why only the $\gamma$-chloroketone product was observed. As illustrated in Fig. 4c, the reactions of each of these three carbon-centred radical species with the four-coordinate Cu(II) complex **I** were thus compared. This led to the formation of the corresponding copper(II)-encounter complexes **XVIIIs$^\alpha$**, **XVIIIs$^\gamma$** and **XVIIIs$^{\gamma c\gamma}$** with respective energies of 1.1, 11.0 and 0.9 kcal/mol. For **XVIIIs$^\alpha$** and **XVIIIs$^\gamma$**, intersystem crossing via the respective minimum energy crossing point species **MECPs$^\alpha$** and **MECPs$^\gamma$** gave the singlet Cu(III) complexes **XIXs$^\alpha$** and **XIXs$^\gamma$**. In contrast, the Cu(II) complex **XVIIIs$^{\gamma c\gamma}$** was found to be a potent reductant and, as such, reduced the metal centre to give the singlet ground-state Cu(I) species **XIXs$^{\gamma c\gamma}$** via minimum energy crossing point species **MECPs$^{\gamma c\gamma}$**. This is followed by the loss of the triflate anion to afford the cationic copper(I) complex **XXs$^{c\gamma}$**, which was found to be exergonic by 3.1 kcal/mol. The greater reactivity of **XVIIs$^{\gamma c\gamma}$** over that of **XVIIs$^\alpha$** or **XVIIs$^\gamma$** toward chlorination was thus delineated to be due to **MECPs$^{\gamma c\gamma}$** (2.6 kcal/mol) lying considerably lower in energy than **MECPs$^\alpha$** (12.5 kcal/mol) or **MECPs$^\gamma$** (16.6 kcal/mol). Once **XXs$^{c\gamma}$** has formed, its chloride ligand acts as a nucleophile and opens the ring of the THFyl motif via S$_N$2 transition state structure **TS$_{XXs}$$^{c\gamma}$$_{-II}$** with an energy barrier of 17.6 kcal/mol in an exergonic fashion with $\Delta G = -7.2$ kcal/mol. The formation of the ensuing C($sp^3$)–Cl bond results in the delivery of the product **2s** along with the regeneration of [Cu(MeCN)$_2$]$^+$ **II**.

For the site-selective chlorination of (*E*)-enones with **3a** as a representative example, HAT reaction at the $\gamma$-position of the substrate by the [TsNCl]• motif was found to be exergonic by 19.0 kcal/mol with an overall energy barrier of 15.2 kcal/mol (Fig. 5a). Producing the alkyl radical species **XXIa** and *p*-TsNHCl, reaction of the former with Cu(II) complex **I** affords Cu(II) adduct **XXIIa** on the triplet surface, which lies 4.1 kcal/mol higher in energy. Subsequently, the minimum energy crossing point species **MECP3a** connects **XXIIa** to organometallic complex **XXIIIa** on the singlet excited-state energy surface in which the copper centre has a formal oxidation state of +1. This change in the oxidation state of the metal centre from +2 to +1 forces the triflate anion to leave to

produce **XXIVa** where the trigonal planar [Cu(MeCN)$_2$Cl] complex is stabilised by the in situ generated carbocation motif. However, we were unable to locate any transition state structures for the Cu–Cl bond forming step. As a consequence, this implied that once **XXIVa** has formed, it immediately undergoes coupling and formation of the C–Cl bond to give the thermodynamically favoured product **4a** and [Cu(MeCN)$_2$]$^+$ **II**, which was found to be exergonic by 15.3 kcal/mol.

In a final set of calculations, the [TsNCl]$^•$ moiety was once again found to serve as a good hydrogen radical atom abstractor in reactions with alkylbenzenes. As depicted in Fig. 5b with **5a** as a representative example, removal of the $\alpha$-H radical atom in the substrate by the [TsNCl]$^•$ motif was shown to require an activation energy of 14.3 kcal/mol and was exergonic by –2.5 kcal/mol. Forming the benzylic radical species **XXVa** and $p$-TsNHCl, the former approaches the tetravalent Cu(II) species **I** with an energy barrier of 2.7 kcal/mol. The thermodynamics of the radical reaction between **XVIIs$^\alpha$** and **5a** to give **1s** and **XXVa** was modelled at the SMD/B3LYP-D3/def2-TZVP//SMD/B3LYP-D3/6-31G($d$,$p$) level of theory to determine if reaction selectivity might be predominantly thermodynamically controlled. The isodesmic reaction was found to be endergonic (by 1.6 kcal/mol, Supplementary Fig. 15), indicating **XVIIs$^\alpha$** is intrinsically more stable than that of **XXVa**. At the same time, our calculations revealed that **XVIIs$^\alpha$** was less reactive toward chlorination with Cu(II) complex **I**, as **MECPs$^\alpha$** is 9.4 kcal/mol higher in energy than **MECP4a** (Figs. 4 and 5). As such, in this case, the intrinsic stability of the radical species is not a determining factor for reactivity. At this juncture, intersystem crossing of the ensuing Cu(I)-encounter complex **XXVIa** on the triplet excited-state energy surface via the minimum energy crossing point species **MECP4a** occurs to furnish the singlet ground-state Cu(III) complex **XXVIIa**. Rapid reductive elimination of the high oxidation state of the metal species leading to C($sp^3$)–Cl bond formation takes place to give benzyl chloride **6a** and [Cu(MeCN)$_2$]$^+$ **II** via **TS$_{XXVIIa-I}$** with an exergonicity of 29.6 kcal/mol.

On the basis of the above experimental and computational results, a tentative mechanism for the Cu(I)-catalysed site-selective C($sp^3$)–Cl bond formation reaction is presented in Fig. 5c. This could involve the initial oxidative addition of dichloramine-T to [Cu(MeCN)$_2$]$^+$ **II**, formed by solvation of CuOTf with acetonitrile, via a SET pathway to give the Cu(II) complex **XII** in which an electron from the metal centre is transferred to the N–Cl bond. As a result, migration of the chlorine atom from the nitrogen centre to that of the metal may occur to give the Cu(II) complex **XIII** where the [TsNCl]$^•$ motif is highly susceptible to dissociation from the metal species. In addition to the formation of the nitrogen-centred radical species, the fragmentation affords the trivalent copper(II) complex **XIV**, which undergoes triflate anion coordination to give the more stable tetravalent Cu(II) species **I**. The removal of an $\alpha$- or $\gamma$-H radical atom in the respective alkylbenzene, ketone and (E)-enone substrate (R–H) by the [TsNCl]$^•$ motif via a HAT pathway might then furnish the R$^•$ species and $p$-tosylchloramide. At this juncture, a divergence in the mode of reactivity is thought to occur depending on the nature of the alkyl radical species (R$^•$). When the R group is that of the ketone (**XVII$^{\gamma cy}$**) or (E)-enone (**XXI**), its potency as a reductant may lead to the unlikely tendency to furnish a stable Cu(III) complex through the formation of a Cu–R bond. As a consequence, reduction of the tetravalent Cu(II) species by the R$^•$ species would deliver the corresponding Cu(I) complexes **XX$^{cy}$** and **XXIV** of the ketone and (E)-enone via an outer-sphere mechanism (Fig. 5c, pathway $a$). Subsequent coupling of the alkyl and chloride groups in the respective Cu(I) complexes would result in C($sp^3$)–Cl bond

formation to give $\gamma$-chloroketone **2** and (E)-$\gamma$-chloroenone **4** along with regenerating [Cu(MeCN)$_2$]$^+$ **II**. In the case of R$^•$ species derived from an alkylbenzene substrate, the decreased reducing ability of the radical adduct might now make the formation of the Cu(III) complex **XXVII** favourable on interacting with Cu(II) species **I** via an inner-sphere mechanism (Fig. 5c, pathway $b$). Reductive elimination in this high oxidation state metal complex would result in C($sp^3$)–Cl bond formation to produce the benzyl chloride **6** and [Cu(MeCN)$_2$]$^+$ **II**. An outer-sphere mechanism might be anticipated to be faster than an inner-sphere one because the two reactive species in the former do not need to be in a direct contact with each other. Consequently, if these two pathways are assumed to be in competition, the redox reaction via the outer-sphere mechanism is predicted to be operative, as evidenced by substrates containing both a benzylic and $\gamma$-C($sp^3$)–H bond where the latter is selectively chlorinated.

In summary, we have elucidated a Cu(I)-catalysed synthetic method for the site-selective chlorination of C($sp^3$)–H bonds of ketones, (E)-enones and alkylbenzenes by dichloramine-T. By realising a rare instance of dichloramine-T as a chlorine radical atom source, it offers a practical and efficient synthetic route to the corresponding chlorinated products with site-selectivities that are among the most selective reported in alkane functionalisation. This site-selectivity was further highlighted by the delivery of only the $\gamma$-chlorinated product in reactions involving ketones and (E)-enones containing a benzylic and/or electron-rich aryl C–H bond. The approach was likewise shown to be chemoselective as the chlorination of unsaturated substrates did not suffer from competitive reactions at the more reactive C=C bond or its transposition. This is often the case in allylic chlorination methods that rely on ene-type reactivity. The key to this product selectivity was suggested to be due to a site-selective HAT reaction between the in situ generated [TsNCl]$^•$ motif and substrate that was dictated by their electronic interactions. Equally crucial was avoiding the intermediacy of the chlorine free radical atom by the formation of the metal-chloride complex to transfer the halogen radical atom to the substrate radical species to give the product. The ability to access various $\gamma$-chloroketones, (E)-$\gamma$-chloroenones and $\alpha$-benzyl chlorides with a single catalytic system will find broad applicability in chemical synthesis and notably impact the way in which many molecules of interest are assembled. In this study, this potential was demonstrated by the late-stage site-selective chlorination of a series of modified bioactive compounds and natural products along with the gram-scale preparation and formal synthesis of two drug molecules. Along with this, we envisage the mechanistic principles provided by our studies will assist future efforts focused on site-selective aliphatic C–H bond functionalisation at other unactivated positions that pervade the compound class.

## Methods

To an oven-dried round-bottom flask (10 mL) was added the ketone **1**, (E)-enone **3** or alkylbenzene **5** (0.2 mmol), CuOTf·0.5PhMe (5.2 mg, 0.02 mmol), NaHCO$_3$ (25.2 mg, 0.3 mmol) and 4 Å MS (50 mg). The reaction vessel was capped and charged with a nitrogen atmosphere through three cycles of the vacuum-nitrogen-backfill method over 10 min. An anhydrous solution of acetonitrile (1 mL) containing dichloramine-T (72 mg, 0.3 mmol for **1** or 57.6 mg, 0.24 mmol **3** and **5**) was then added and the resulting reaction mixture was stirred at room temperature for 24 h. For substrates **1s–x**, a further amount of CuOTf·0.5PhMe (5.2 mg, 0.02 mmol) and dichloramine-T (72 mg, 0.3 mmol) was added at this point and the reaction was stirred for an additional 24 h. Upon completion, the reaction mixture was filtered through a Celite pad with dichloromethane (3 × 1 mL). On removing the organic solvent under reduced pressure, the resulting residue was purified by flash column chromatography on silica gel (petroleum ether/diethyl ether as eluent) to give the desired chlorination product.

## Data availability

The data supporting the findings of this study are available in the paper and its Supplementary Information; further data are available from the corresponding author on request. The X-ray crystallographic coordinates for structures reported in this study have been deposited at the Cambridge Crystallographic Data Centre (CCDC), under deposition numbers CCDC 1562076 (**2d**) and CCDC 1562075 (**4k**). These data can be obtained free of charge from the Cambridge Crystallographic Data Centre via www.ccdc.cam.ac.uk/data_request/cif.

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

## Acknowledgements
This work was supported by a Discovery Project Grant (DP210103425) from the Australian Research Council.

## Author contributions
P.W.H.C. conceived and directed the project. P.W.H.C., J.J., Y.Z. and S.H.K. designed the experiments and analysed the data. J.J. and Y.Z. performed all of the experiments. A.A. and K.F. performed all of the computational studies. All the authors contributed to the preparation and writing of the manuscript.

## Competing interests
The authors declare no competing interests.
