## [Peer Review File · Nature Communications]

Reviewers' Comments:

Reviewer #1:

Remarks to the Author:

This paper reports on a Cu-catalyzed sp³ CH bond chlorination. The results are interesting but the interpretation is not consistent. As it stands now, I see a thermodynamically controlled reaction but no evidence for the claimed site-selectivity that would imply kinetic control (any selectivity must be kinetic beyond intrinsic thermodynamic preference). Also, the computations are not adequate for this system.

In order of appearance, these are my comments:

I often read "mild conditions" – what defines mild conditions? This is probably quite subjective so I would refrain from such attributes.

As there is (to the best of my knowledge) no intramolecular aromatic chlorination, the notion of "intermolecular" is not sensible.

The way the reaction equations are drawn is extremely confusing. For instance, I assumed that in Fig 1d all four starting materials are reagents in a competitive reaction to assess the relative rates of reactions. That is, as I understood from analyzing Fig 2, not the case. I've never see such an odd presentation and cannot accept this way of writing chemical reactions. Please write them the normal way to avoid this confusion.

Yield is the amount of *pure* product in moles relative to the moles of starting material. An NMR "yield" is not a yield. Think of having to make a chlorinated drug: only a pure product with a proper yield would have a market and an application. So "NMR yields" are ok in an exploratory table but in the text it must also be marked as such. "Product yield" is redundant, "yield" will do.

The efficacy of a catalyzed reaction is measured in TON and even better in TOF numbers. These should be given so an industrial chemist can decide whether this reaction is worth for a large-scale process. Currently the TOFs are in the range of 0.05–0.2 per hour; this is not an efficient catalytic process.

What is the role of molecular sieves? Water should have little to no effect on the reaction. Or does it? What happens without 4 Å MS?

Fig 2: It looks very much like the selectivity is entirely covered by the thermodynamic stability of the incipient radical that is then chlorinated; the authors convincingly argue that they have observed a radical mechanism. Have the authors computed these radicals to determine the thermodynamics? If selectivity is claimed, the difference must be in competing transition structures of kinetically controlled reaction pathways, otherwise there is no selectivity, just thermodynamics. This is still interesting but no control over the catalytic process can be exerted in terms of selectivity, and that is important. Note that polar effects on radicals have significant consequences for BDEs that are not a good measure of reactivity in polar radical reactions. One can evaluate the radical thermodynamics computationally usingisodesmic equations.

Figure 3g implies that the KIE might be large. What is it? Is it beyond the classical limit at this temperature (about 7–8)? If so, quantum mechanical tunneling (common for H-transfer) must be involved and this must be mentioned.

The theory level given as "B3LYP-D3-SMD/6-31G(d),LanL2DZ//B3LYP-D3-SMD/def2-TZVP" is confusing because it means that the level for optimization (B3LYP-D3-SMD/def2-TZVP) is higher than the level for energy evaluation (B3LYP-D3-SMD/6-31G(d),LanL2DZ). This is how this is to be read. Clarify. In either case, the level of theory is not meaningful as polarization is missing on all heavy atoms. This is particularly important for Cu; the LanL2DZ basis sets must be amended or replaced by much better and more modern basis sets. Have the authors anchored their computations to some experimental value? The KIE would be optimal for this as this involved computing the TSs for the rate-limiting step. The text should also provide how spin and charge

densities were determined. NBO? That would be a sensible choice for the charges.

Reviewer #2:

Remarks to the Author:

This manuscript describes the development of a copper-catalyzed method for the direct chlorination of C(sp³)-H bonds of ketone-containing and benzylic substrates. To develop this method, the authors capitalized on a previously-overlooked observation that dichloramine-T, a reagent commonly used for N-transfer reaction, can actually be used as a chlorinating agent. Examination of the use of this reagent in combination with transition metal catalysts revealed that a system containing Cu(OTf) and TsNCl₂ can catalyze selective gamma functionalization of linear ketone substrates. More thorough substrate scope examination revealed that such gamma selectivity is maintained with other ketone-containing substrates as well as enone substrates. Furthermore, alkylbenzenes can be chlorinated at the benzylic position and the method can be used for the functionalization of natural products and drug molecules. The authors then performed a series of mechanistic studies using substrate probes, and complemented this study with DFT calculation. These combined studies suggest that a combination of Cu(OTf) and TsNCl₂ can lead to the generation of an N-centered radical species and a Cu(II) species containing Cl ligand. The former performs hydrogen atom abstraction on the substrate and the regioselectivity is dictated by the electronic bias introduced by the presence of the ketone group. The resulting radical can then react with the Cu(II) species, leading to the chlorination product. To the best of my knowledge, the development of copper-based system for C(sp³)-H chlorination is still pretty rare and the use of TsNCl₂ for such purpose is not very common. I also think that this work could inspire the development of related Cu-based strategies for other types of C-H functionalization. For these reasons, I believe that the work is suitable for publication in Nature Communications. Before acceptance however, I have several comments/concerns that should be addressed by the authors, as listed below:

- The radical clock experiment in figure 3B is quite odd to me. Firstly, the authors were not able to isolate the product from the reaction and the product identity was only assigned from crude H NMR and HRMS. The crude NMR looks quite messy, how can the authors confidently deduce that the primary chloride (compound 15) is the product structure? Secondly, if the reaction proceeds via radical reaction, it seems that the pathway to product compound 15 will have to go through some form of vinylcyclopropane rearrangement. More detailed discussion should be provided as well as some form of reconciliation between this result and the DFT results.
- It seems odd to me that compound 18 failed to give any chlorination product while compounds 10 and 11 were chlorinated successfully. The only main difference is that the latter 2 contain an EWG ester. Even then, these esters are rather far away from the reaction site and I would think that the electronic influence would be quite minimal. Can the authors comment on this?
- In Fig 1B, the authors showed Alexanian's work whereby other regioisomers of the chlorination product were observed. At the same time, the text mentioned the generally poor site-selectivity of current C-H chlorination methods. I think it would only be fair if the authors tested the same substrate in fig 1B in chlorination with their method as a benchmark in a head-to-head comparison. In the same vein, I noticed that virtually all the substrates provided in Fig 2 contains an aromatic group, even the ester tested is a phenoxy ester. What are the outcomes of the reaction when aliphatic esters or amides are used as substrate? It seems that the proposed mechanism shouldn't preclude the use of these substrates.

Reviewer #3:

Remarks to the Author:

I have been explicitly asked to review the computational section of this work, hence my comments will only cover this.

The authors present an extensive DFT study of possible mechanistic scenarios for the chlorination reaction. The computations appear to have been conducted meticulously and a number of alternative pathways and species have been studied to give a rather complete picture of thermodynamic and kinetic stabilities. The proposed mechanistic cycle appears credible and is well

supported, in particular by the DFT calculations, but are also consistent with experimental observations.

The only major issue I have with the work as it stands is that the computational methodology is rather weakly justified. I could not find a clear, explicit, referenced justification for choice of functionals, basis sets and the effective core potential. I mean, LANL2DZ is a very old ECP that has been actively used since the 90s at least. There must be some developments in this field, I would assume. In any case, there should be explicit justification for all the elements in the computational methods.

Recommendation: accept after addressing the issues above

Point-by-Point Response to the Reviewers' Comments

Reviewer #1

1. The reviewer notes: "I often read "mild conditions" – what defines mild conditions? This is probably quite subjective so I would refrain from such attributes."

We would like to thank the reviewer's suggestion to remove the phrase "mild conditions" as it is subjective. We have replaced the phrase with "room temperature" in the Abstract on page 1 and removed it from the discussion on page 5 of the revised manuscript.

2. The reviewer notes: "As there is (to the best of my knowledge) no intramolecular aromatic chlorination, the notion of "intermolecular" is not sensible."

We would like to thank the reviewer for pointing out that as there are no intramolecular aromatic chlorination methods, it is unnecessary to use the adjective "intermolecular". This has been removed from the discussion on page 2 of the revised manuscript.

3. The reviewer notes: "The way the reaction equations are drawn is extremely confusing. For instance, I assumed that in Fig 1d all four starting materials are reagents in a competitive reaction to assess the relative rates of reactions. That is, as I understood from analyzing Fig 2, not the case. I've never see such an odd presentation and cannot accept this way of writing chemical reactions. Please write them the normal way to avoid this confusion."

We would like to apologise if the reviewer found Fig 1d difficult to follow and extremely confusing. Fig 1d has been modified so that it avoids the possible interpretation that all four starting materials are reagents in a competitive reaction to determine the relative rates of reactions.

4. The reviewer notes: "Yield is the amount of *pure* product in moles relative to the moles of starting material. An NMR "yield" is not a yield. Think of having to make a chlorinated drug: only a pure product with a proper yield would have a market and an application. So "NMR yields" are ok in an exploratory table but in the text it must also be marked as such. "Product yield" is redundant, "yield" will do."

We would like to thank the reviewer's suggestion to replace the term "product yield" with "yield" or "¹H NMR yield" in the text where appropriate. We have made these changes in the discussion on pages 6–9 of the revised manuscript.

5. The reviewer notes: "The efficacy of a catalyzed reaction is measured in TON and even better in TOF numbers. These should be given so an industrial chemist can decide whether this reaction is worth for a large-scale process. Currently the TOFs are in the range of 0.05–0.2 per hour; this is not an efficient catalytic process."

We would like to thank the reviewer for their suggestion to provide details of this aspect of our Cu(I)-catalysed chlorination method. Because our current study is focused on reaction discovery, we did not optimise the reaction with either TON/TOF in mind, and therefore feel that it might be misleading to report these values in the present work. On the other hand, as part of future studies in this field of copper catalysis, we would like to also thank the reviewer for prompting us to include work that will focus on optimising the most industrially relevant reactions with high TON/TOF values.

6. The reviewer notes: “What is the role of molecular sieves? Water should have little to no effect on the reaction. Or does it? What happens without 4 Å MS?”.

We would like to thank the reviewer for their query and the suggested control experiment. The role of 4 Å MS is to minimise the trace amounts of water that remain present in distilled acetonitrile as dichloramine-T is moisture sensitive. This proposed role of 4 Å MS is supported by our findings from the control experiment suggested by the reviewer showing the CuOTf-catalysed chlorination of **1a** by dichloramine-T under the optimised reaction conditions and in the absence of 4 Å MS gave **2a** in 48% yield. In comparison and as noted in the discussion and in Table 1, entry 1 of the revised manuscript, we had previously found the analogous reaction with 4 Å MS afforded **2a** in 78% yield. This new result has been added to the discussion on page 9 of the revised manuscript and as entry 17 in Table S1 on page S8 of the Supplementary Information.

7. The reviewer notes: “Fig 2: It looks very much like the selectivity is entirely covered by the thermodynamic stability of the incipient radical that is then chlorinated; the authors convincingly argue that they have observed a radical mechanism. Have the authors computed these radicals to determine the thermodynamics? If selectivity is claimed, the difference must be in competing transition structures of kinetically controlled reaction pathways, otherwise there is no selectivity, just thermodynamics. This is still interesting but no control over the catalytic process can be exerted in terms of selectivity, and that is important. Note that polar effects on radicals have significant consequences for BDEs that are not a good measure of reactivity in polar radical reactions. One can evaluate the radical thermodynamics computationally using isodesmic equations.”.

We would like to thank the reviewer for the suggestion to computationally determine if reaction selectivity might be predominantly thermodynamically controlled. In response, the thermodynamics of the radical reaction between **XVII^α** and **5a** to give **1s** and **XXVa** was modelled at the SMD/B3LYP-D3/def2-TZVP//SMD/B3LYP-D3/6-31G(d,p) level of theory. The isodesmic reaction was found to be endergonic (by 1.6 kcal/mol), indicating **XVII^α** is intrinsically more stable than that of **XXVa**. At the same time, our calculations revealed that **XVII^α** was less reactive toward chlorination with Cu(II) complex **I**, as **MECPs^α** is 9.4 kcal/mol higher in energy than **MECP4a**, as shown in Figs. 4 and 5 of the revised manuscript. As such, in this case, the intrinsic stability of the radical species is not a determining factor for reactivity. We have added the findings of the isodesmic reaction to the discussion on page 22 of the revised manuscript as a reference footnote (ref. 56) and as Supplementary Fig. 14 on page S74 of the Supplementary Information.

8. The reviewer notes: “Figure 3g implies that the KIE might be large. What is it? Is it beyond the classical limit at this temperature (about 7–8)? If so, quantum mechanical tunneling (common for H-transfer) must be involved and this must be mentioned.”.

We would like to thank the reviewer for highlighting the possible involvement of quantum mechanical tunnelling and, if this is the case, it should be noted. We agree with the reviewer that quantum mechanical tunnelling should be operative in our study and to confirm this to be the case, the Cu(I)-catalysed in parallel and competitive KIE control experiments of **5c** and **d₃-5c** were repeated at 50 °C and found to give respective k_H/k_D values of 6.0 and 5.8. The theoretical limit for k_H/k_D (at 298 K) of 6.5–7 is estimated for complete C–H bond breaking in H[•] atom transfer reactions (Gómez-Gallego, M. & Sierra, M. A. Kinetic isotope effects in the study of organometallic reaction mechanisms. *Chem. Rev.* **111**, 4857–4963 (2011)). A large KIE value, like that measured in our study, is considered an experimental indicator that quantum mechanical tunnelling may be occurring. This is further supported by our

computational findings, showing that the KIE value for H[•] atom abstraction from **5c** is 6.4 which is close to the theoretical limit and corresponds with our experimentally observed results. We have added these new findings to the discussion on page 14 of the revised manuscript as a reference footnote (ref. 55), updated Scheme 3g and experimental section on pages S18–S22 and Supplementary Fig. 6 on page S22 of the Supplementary Information. The spectroscopic data and ¹H and ¹³C NMR spectra of *d*₂-**6c** has also been added to the Supplementary Information on pages S69 and S396, respectively.

9. The reviewer notes: “The theory level given as “B3LYP-D3-SMD/6-31G(d),LanL2DZ//B3LYP-D3-SMD/def2-TZVP” is confusing because it means that the level for optimization (B3LYP-D3-SMD/def2-TZVP) is higher than the level for energy evaluation (B3LYP-D3-SMD/6-31G(d),LanL2DZ). This is how this is to be read. Clarify.

We would like to thank the reviewer for highlighting this typing error, which has been changed to correctly read: SMD/B3LYP-D3/def2-TZVP//SMD/B3LYP-D3/6-31G(d,p),LANL2DZ(f).

10. The reviewer notes: “In either case, the level of theory is not meaningful as polarization is missing on all heavy atoms. This is particularly important for Cu; the LanL2DZ basis sets must be amended or replaced by much better and more modern basis sets.”.

We would like to thank the reviewer for highlighting these aspects of our DFT calculations. In response, we would have made the following revisions:

(i) For the noted missing polarisation on all heavy atoms, we have addressed this point, as described in the computational details in the Supplementary Information, by carrying out the single point calculations with the large basis set, def2-TZVP. This basis set considers the f polarisation function of exponent 2.233 for the copper atom. We have also added the f polarisation function of exponent 3.525 for the copper atom to the LANL2DZ basis set calculations which are provided in the Supplementary Information for comparison.

(ii) Following the comment that “the LanL2DZ basis sets must be amended or replaced by much better and more modern basis sets.”, as described in the computational details, we have replaced the LANL2DZ basis set for the copper with the more modern SDD basis set with effective core potential, and for all remaining atoms replaced 6-31G(d) with the larger 6-31G(d,p) basis set. The SMD/B3LYP-D3/def2-TZVP//SMD/B3LYP-D3/6-31G(d,p),SDD level of theory aligns with recently reported computational studies on copper mediated systems listed below and cited in the Supplementary Information as references S30–S40.

(a) Basumatary, B., Hashiguchi, I., Mori, S., Shimizu, S., Ishida, M. & Furuta, H. Copper 1,19-diaza-21,24-dicarbacorrole: A corrole analogue with an N–N linkage stabilizes a ground-state singlet organocopper species. *Angew. Chem.* **132**, 16031–16035 (2020); (b) Hu, H., Chen, S. J., Mandal, M., Pratik, S. M., Buss, J. A., Krska, S. W., Cramer, C. J. & Stahl, S. S. Copper-catalysed benzylic C–H coupling with alcohols *via* radical relay enabled by redox buffering. *Nat. Catal.* **3**, 358–367 (2020); (c) Paradisi, A., Johnston, E. M., Tovborg, M., Nicoll, C. R., Ciano, L., Dowle, A., McMaster, J., Hancock, Y., Davies, G. J. & Walton, P. H. Formation of a copper(II)–tyrosyl complex at the active site of lytic polysaccharide monoxygenases following oxidation by H₂O₂. *J. Am. Chem. Soc.* **141**, 18585–18599 (2019); (d) Mandal, M., Elwell, C. E., Bouchey, C. J., Zerk, T. J., Tolman, W. B. & Cramer, C. J. Mechanisms for hydrogen-atom abstraction by mononuclear copper(III) cores: hydrogen-atom transfer or concerted proton-coupled electron transfer? *J. Am. Chem. Soc.* **141**, 17236–17244 (2019); (e) Jiao, Y., Chiou, M. F., Li, Y. & Bao, H. Copper-catalyzed radical acyl-

cyanation of alkenes with mechanistic studies on the *tert*-butoxy radical. *ACS Catal.* **9**, 5191–5197 (2019); (f) Isegawa, M., Sameera, W. M. C., Sharma, A. K., Kitano, T., Kato, M., Kobayashi, S. & Morokuma, K. Copper-catalyzed enantioselective boron conjugate addition: DFT and AFIR study on different selectivities of Cu(I) and Cu(II) catalysts. *ACS Catal.* **7**, 5370–5380 (2017); (g) Collins, L. R., Rajabi, N. A., Macgregor, S. A., Mahon, M. F. & Whittlesey, M. K. Experimental and computational studies of the copper borate complexes [(NHC)Cu(HBEt₃)] and [(NHC)Cu(HB(C₆F₅)₃)]. *Angew. Chem.* **128**, 15768–15772 (2016); (h) Lemon, C. M., Huynh, M., Maher, A. G., Anderson, B. L., Bloch, E. D., Powers, D. C. & Nocera, D. G. Electronic structure of copper corroles. *Angew. Chem.* **128**, 2216–2220 (2016); (i) Stopka, T., Marzo, L., Zurro, M., Janich, S., Wuerthwein, E. U., Daniliuc, C. G., Aleman, J. & Mancheno, O. G. Oxidative C–H bond functionalization and ring expansion with TMSCHN₂: A copper(I)-catalyzed approach to dibenzoxepines and dibenzoazepines. *Angew. Chem. Int. Ed.* **54**, 5049–5053 (2015); (j) Ling, L., Liu, K., Li, X. & Li, Y. General reaction mode of hypervalent iodine trifluoromethylation reagent: A density functional theory study. *ACS Catal.* **5**, 2458–2468 (2015); (k) Zhao, G. M., Liu, H. L., Zhang, D. D., Huang, X. R. & Yang, X. DFT study on mechanism of *N*-alkylation of amino derivatives with primary alcohols catalyzed by copper(II) acetate. *ACS Catal.* **4**, 2231–2240 (2014).

The results obtained at the SMD/B3LYP-D3/def2-TZVP//SMD/B3LYP-D3/6-31G(d,p),SDD level of theory have been added to the discussion on pages 15–22 of the revised manuscript. We have also modified Figs. 4 and 5 on pages 16 and 21 of the revised manuscript and Supplementary Figs. S11–S14 on pages S73 and S74 of the Supplementary Information (with raw data presented in Supplementary Table 2 on pages S74–S177) accordingly. Our previous calculations at the SMD/B3LYP-D3/def2-TZVP//SMD/B3LYP-D3/6-31G(d),LANL2DZ(f) level of theory have been placed in the Supplementary Information as Supplementary Figs S15–22 on pages S177–S183 (with raw data presented in Supplementary Table 3 on pages S183–S287) for comparison.

11. The reviewer asks: “Have the authors anchored their computations to some experimental value?”.

We aligned our computational studies on the related recently reported computational studies on copper mediated systems listed in our reply to Q10 and cited in the Supplementary Information as references S30–S40.

12. The reviewer notes: “The text should also provide how spin and charge densities were determined. NBO? That would be a sensible choice for the charges.”.

We would like to thank the reviewer for suggesting the way in which the spin and charge densities were determined should be mentioned in the text. To address this point, we have added the following sentence to the computational details on page S71 in the Supplementary Information: Spin density distributions were obtained by Mulliken population analysis using the SMD/B3LYP-D3/def2-TZVP//BS1 level of theory in acetonitrile.

Reviewer #2

13. The reviewer notes: “The radical clock experiment in figure 3B is quite odd to me. Firstly, the authors were not able to isolate the product from the reaction and the product identity was only assigned from crude ¹H NMR and HRMS. The crude NMR looks quite messy, how can the authors confidently deduce that the primary chloride (compound **15**) is the product structure? Secondly, if the reaction proceeds via radical reaction, it seems that the pathway to product compound **15** will have to go through some form of vinylcyclopropane rearrangement. More detailed discussion should be provided as well as some form of reconciliation between this result and the DFT results.”

We would like to thank the reviewer for highlighting the drawing error in Fig. 3b. This has been corrected with the primary structure of the 1,3-chlorodiene **15** replaced with one that represents a mixture of regioisomers of the product. The reviewer is correct to note that the assignment of the 1,3-chlorodiene **15** as the primary adduct cannot be definitively deduced based solely on its crude ¹H NMR spectrum. We were only able to detect the presence of the 1,3-chlorodiene **15** in the crude reaction mixture by HRMS measurements. The intention of providing the crude ¹H NMR spectrum of the Cu(I)-catalysed reaction of **31** with dichloramine-T in the Supplementary Information was to show the complete consumption of the substrate and provide indirect further support for the proposed radical mechanism. We would like to apologise for any confusion that this, in combination with the drawing error, might have caused.

We would also like to thank the reviewer for the suggestion that the reaction of **31** might proceed *via* a vinylcyclopropane-type rearrangement pathway. We agree that this is likely to be the case and, while the mechanism currently remains unclear, we speculate that a vinylcyclopropane-type rearrangement pathway might be initiated by γ -H[•] atom abstraction of the substrate as suggested by our DFT calculations. The resulting diradical species might subsequently undergo dichlorination with the proposed *in situ* formed copper(II) complex **I**. We hypothesize dehydrohalogenation of the ensuing dechlorinated adduct might then provide 1,3-chlorodiene **15** as a mixture of regioisomers. We have added a concise version of this proposed pathway to the discussion on page 13 of the revised manuscript as a reference footnote (ref. 54).

14. The reviewer notes: “It seems odd to me that compound **18** failed to give any chlorination product while compounds **10** and **11** were chlorinated successfully. The only main difference is that the latter 2 contain an EWG ester. Even then, these esters are rather far away from the reaction site and I would think that the electronic influence would be quite minimal. Can the authors comment on this?”

We would like to thank the reviewer for highlighting this oversight and apologise for any confusion this may have caused. The α -chlorinated product **6m** is afforded in 70% yield in the control experiment described in Fig. 3e but the reporting of this result had been inadvertently overlooked due to our eagerness to note that the γ -chloroketone (renumbered as compound **16** in the revised manuscript) was not obtained. This missing result has been added to the discussion on page 13 of the revised manuscript and Fig. 3e has been updated. The spectroscopic data and ¹H and ¹³C NMR spectra of **6m** has also been added to the Supplementary Information on pages S65 and S386, respectively.

15. The reviewer notes: “In Fig 1B, the authors showed Alexanian’s work whereby other regioisomers of the chlorination product were observed. At the same time, the text mentioned the generally poor site-selectivity of current C-H chlorination methods. I think it would only be fair if the authors tested the same substrate in fig 1B in chlorination with their method as a benchmark in a head-to-head comparison.”

We would like to thank the reviewer for the suggestion to benchmark our Cu(I)-catalysed chlorination method with the reaction of methyl hexanoate that was reported by the Alexanian group. The reference compound methyl 5-chlorohexanoate was prepared following the procedure reported by Alexanian and co-workers. However, the reported GC retention time for methyl 5-chlorohexanoate could not be reproduced as only the identity of the GC gas chromatograph and column was provided. As shown in Figs. 1 and 2, retention times of $t_R = 16.208$ and 16.266 min for the reference compound was subsequently determined by GC analysis and ^1H NMR measurements before and after its purification by flash column chromatography on silica gel. The ^1H NMR spectral data of methyl 5-chlorohexanoate were in accordance with that reported by Henry and co-workers (Hamed, O., El-Qisairi, A. & Henry, P. M. Palladium(II)-catalyzed oxidation of aldehydes and ketones. 1. Carbonylation of ketones with carbon monoxide catalyzed by palladium(II) chloride in methanol. *J. Org. Chem.* **66**, 180–185 (2001)). We next performed the reaction of methyl hexanoate with our Cu(I)-catalysed chlorination method with dichloramine-T. As shown in Fig. 2, this revealed no chlorination adducts of the methyl ester were detected by either GC analysis or ^1H NMR measurements of the crude mixture and only a mixture of unidentifiable decomposition products were obtained. The reaction was repeated a further two times to confirm that our Cu(I)-catalysed method for the site-selective chlorination of ketones, (*E*)-enones and alkylbenzenes by dichloramine-T was, unfortunately, not applicable to carboxylic esters containing a secondary $\gamma\text{-C}(\text{sp}^3)\text{-H}$ bond. This new result has been added to the discussion on page 9 and Fig. 2 on page 10 of the revised manuscript.

Fig. 1 GC spectra of (a) methyl hexanoate, (b) crude reaction mixture of methyl 5-chlorohexanoate, (c) methyl 5-chlorohexanoate after purification by flash column chromatography on silica gel, and (d) crude reaction mixture of the Cu(I)-catalysed reaction of methyl hexanoate and dichloramine-T.

Fig. 2 ^1H NMR spectra of (a) methyl hexanoate, (b) crude reaction mixture of methyl 5-chlorohexanoate, (c) methyl 5-chlorohexanoate after purification by flash column chromatography on silica gel, and (d) crude reaction mixture of the Cu(I)-catalysed reaction of methyl hexanoate and dichloramine-T.

16. The reviewer notes: “In the same vein, I noticed that virtually all the substrates provided in Fig 2 contains an aromatic group, even the ester tested is a phenoxy ester. What are the outcomes of the reaction when aliphatic esters or amides are used as substrate? It seems that the proposed mechanism shouldn’t preclude the use of these substrates.”.

We would like to thank the reviewer for their query and the suggested control experiments with aliphatic esters or amides as the substrate. The reviewer correctly concludes that the proposed mechanism of our Cu(I)-catalysed chlorination method should also be applicable to substrates that do not contain an aromatic group. The inclusion of an aromatic group in the substrates examined in Fig. 2 of the manuscript was to enable the detection of the chlorinated product by TLC analysis with visualisation by UV light (254 nm) during purification by flash column chromatography on silica gel. In the absence of an aryl group, TLC analysis of the crude reaction mixtures of the ketone **1**, and carboxylic ester **2** and amide **3** with a wide variety of TLC stains was found to be problematic (Fig. 3).

Fig. 3 Aliphatic ketone (**1**), carboxylic ester (**2**) and amide (**3**) chosen to show the influence of an absent aryl group on the Cu(I)-catalysed chlorination method.

For reaction of the ketone **1**, we found visualisation of the chlorinated product could only be achieved with vanillin. Without the possibility to perform visualisation by UV light, purification by flash column chromatography on silica gel of the crude mixture obtained from the reaction of the cyclopentyl-substituted ketone **1** had to be carefully repeated 4 times before ^1H NMR measurements indicated that the product was isolated as a single compound in 45% yield (Fig. 4). However, ^{13}C NMR measurements subsequently showed that this was not the case and a very small amount of one or more unknown compounds were also present, which were not UV light-active nor detectable with various TLC stains. For reactions involving the carboxylic ester **2** and amide **3**, visualisation could not be achieved with various TLC stains such as ninhydrin and hydroxylamine/ FeCl_3 . For this reason, purification by flash column chromatography on silica gel of the crude mixtures was not attempted.

Fig. 4 ^1H and ^{13}C NMR spectra of 4-chloro-1-cyclopentyl-4-methylpentan-1-one.

To demonstrate that an aryl group has no effect on the outcome of our Cu(I)-catalysed chlorination method, the reaction of the *N*-phthalimidyl-substituted carboxylic ester shown in Fig. 5 was examined. This revealed the UV light-active γ -chlorinated product could be readily isolated in 50% yield by flash column chromatography on silica gel. This new result has been added to the discussion on page 9 and Fig. 2 on page 10 of the revised manuscript as substrate **1z** and product **2z**. The spectroscopic data and ^1H and ^{13}C NMR spectra of the substrate **1z** and product **2z** has also been added to the Supplementary Information on pages S35, S55, S310 and S361, respectively.

Fig. 5 Cu(I)-catalysed site-selective C(sp³)-H bond chlorination of carboxylic ester **1z** by dichloramine-T.

Reviewer #3

17. The reviewer notes: “The only major issue I have with the work as it stands is that the computational methodology is rather weakly justified. I could not find a clear, explicit, referenced justification for choice of functionals, basis sets and the effective core potential. I mean, LANL2DZ is a very old ECP that has been actively used since the 90s at least. There must be some developments in this field, I would assume. In any case, there should be explicit justification for all the elements in the computational methods.”.

We would like to thank the reviewer for the suggestion and would like to refer them to our reply to Q10 from Reviewer #1, which addressed these concerns.

Reviewers' Comments:

Reviewer #1:

Remarks to the Author:

The paper has been improved considerably in the revision stage and can be published provided the following aspects are addressed:

The authors try to avoid the TON/TOF discussion by saying that their reactions were not optimized while the paper says the opposite: The caption of Table 1 reads quite clearly: "Optimisation of the reaction conditions". Note that in their rebuttal the same happens when the authors state "under the optimised reaction conditions" in the paragraph after saying "we did not optimise the reaction with either TON/TOF in mind". Of course, there is only one type of optimization in this type of chemistry, namely maximizing the yield over time. Hence, the authors should give, as it is necessary to gauge the efficacy of the catalytic process, TOF-numbers. These can be calculated in minutes by hand and readily be added in an extra column in Table 1 and next to the yields in Figure 2.

I remain unconvinced about the kinetic vs. thermodynamic situation of the present reaction. Theisodesmic evaluation of just one example is within chemical error, so it is hard to tell. But this will perhaps have to await a more detailed mechanistic study.

A pertinent current reference to the involvement of quantum tunneling in reaction mechanisms may be Trends Chem. 2020, 2, 980.

As a (somewhat funny) side note: Yields cannot be isolated, only products, so "isolated yield" is not meaningful even when it is used a lot, it is still wrong.

Peter R. Schreiner

Reviewer #2:

Remarks to the Author:

All of my concerns have been addressed by the authors in the revised manuscript. I believe that the quality of the manuscript has been much improved during the revision and it is now suitable for publication in Nature Communications.

Reviewer #3:

Remarks to the Author:

my concerns have been addressed.

Point-by-Point Response to the Reviewers' Comments

Reviewer #1

1. The reviewer notes: "The authors try to avoid the TON/TOF discussion by saying that their reactions were not optimized while the paper says the opposite: The caption of Table 1 reads quite clearly: "Optimisation of the reaction conditions". Note that in their rebuttal the same happens when the authors state "under the optimised reaction conditions" in the paragraph after saying "we did not optimise the reaction with either TON/TOF in mind". Of course, there is only one type of optimization in this type of chemistry, namely maximizing the yield over time. Hence, the authors should give, as it is necessary to gauge the efficacy of the catalytic process, TOF-numbers. These can be calculated in minutes by hand and readily be added in an extra column in Table 1 and next to the yields in Figure 2."

We would like to once again thank the reviewer's suggestion to provide details of this aspect of our Cu(I)-catalysed chlorination method. However, we have not added the TOF numbers of our catalytic system to the manuscript as we would like to avoid giving the reader the wrong impression that it is a potentially viable industrial process and thus, one of the highlights of our study. We would like to highlight to the reader that the novelty of our study is the site-selective chlorination of ketones, (*E*)-enones and alkylbenzenes by dichloramine-T to give the corresponding γ -chloroketones, (*E*)- γ -chloroenones and α -benzyl chlorides with a single catalytic system. It is not our intention to highlight the efficacy of the reaction, which, as noted by the reviewer, gave TOF numbers in the range of 0.05–0.40 h⁻¹ and if this was added to the manuscript, we believe it will distract the reader from our aforementioned aim. We would also like to add that, while the reviewer is correct in noting that the TOF numbers can be calculated in minutes by hand and readily added to Table 1 as an extra column, unfortunately, this is not the case for Figure 2. On attempting to add the TOF numbers to Figure 2, we found this was not possible in a number of places due to the limited amount of available space. As a result, this made Figure 2 look messy and difficult to read. In attempting to resolve this problem by creating more space, we found that the schematic of Figure 2 took up almost all of the page, which we believe will not leave enough room for the Figure 2 legend when typeset into the template of the journal. That said, we do agree with the reviewer that the TOF numbers give chemists with an easy means to gauge the efficacy of our catalytic method and a starting point to deciding whether it is worth developing it into a large-scale process. With this in mind, we have, therefore, added these values to the Compound Characterisation Data section of the Supplementary Information.

2. The reviewer notes: "A pertinent current reference to the involvement of quantum tunneling in reaction mechanisms may be Trends Chem. 2020, 2, 980."

We would like to thank the reviewer for the suggested reference, which has been added to the reference section of the manuscript as reference 59.

Reviewer #2

3. No further revisions were requested by Reviewer #2.

Reviewer #3

4. No further revisions were requested by Reviewer #3.